# The Role of Soluble Corn Fiber on Glycemic and Insulin Response

**DOI:** 10.3390/nu12040961

**Published:** 2020-03-30

**Authors:** Wei Shuan Kimberly Tan, Pei Fen Winnie Chia, Shalini Ponnalagu, Kavita Karnik, Christiani Jeyakumar Henry

**Affiliations:** 1Clinical Nutrition Research Centre (CNRC), Singapore Institute for Clinical Sciences (SICS), Agency for Science, Technology and Research (A*STAR) and National University Health System, Singapore 117599, Singapore; Kimberly_Tan@hq.a-star.edu.sg (W.S.K.T.); chiawinniechia@gmail.com (P.F.W.C.); Shalini_Ponnalagu@sics.a-star.edu.sg (S.P.); 2Innovation and Commercial Development, Tate & Lyle Ingredients Americans LLC, Hoffman Estates, IL 60192, USA; Kavita.Karnik@tateandlyle.com; 3Department of Biochemistry, Yong Loo Lin School of Medicine, National University of Singapore, Singapore 117596, Singapore

**Keywords:** glycaemic response, glucose, insulin, soluble dietary fiber

## Abstract

Increasing prevalence of type 2 diabetes mellitus (T2DM) in Asia has prompted the exploration of dietary fibers as an ingredient to attenuate glycemic response (GR). This study aims to compare the effects of replacing 50% of total carbohydrate with soluble corn fiber (SCF) or maltodextrin on the GR and insulin response (IR). In this randomized cross-over study, twenty-two healthy Chinese males aged between 21–60 years were recruited. The participants consumed glucose beverages and four test meals comprising SCF or maltodextrin in glutinous rice or as a drink. Repeated-measure ANOVA was used to compare the incremental area under the curve values of glucose (iAUGC) and insulin (iAUIC) of all the foods. Relative response (RR) of the beverages were also calculated and compared using paired t-test. SCF treatments had significantly lower iAUGC (*p*-value < 0.05) and iAUIC (*p*-value < 0.001) as compared to all treatments. Both treatments (rice and beverage) of maltodextrin were not significantly different from glucose (*p*-value > 0.05). Maltodextrin beverage had significantly increased postprandial GR and insulin secretion by 20% and 40%, respectively, when compared to SCF beverage (*p*-value < 0.001). This study shows that the inclusion of SCF into the diet is beneficial in controlling the postprandial GR. Replacing total carbohydrates with SCF effectively lowers GR and IR.

## 1. Introduction

The prevalence of type 2 diabetes mellitus (T2DM) continues to increase globally at an alarming rate especially in Asia [1]. Carbohydrate metabolism may play an important role in the development of T2DM [2]. The quality of carbohydrates has received particular interest in its ability to be manipulated either by processing or the addition of ingredients to alter carbohydrate breakdown, absorption, or utilization [3]. This has enabled food industries to develop a range of foods that can attenuate blood glucose response and to influence insulin sensitivity [3]. Examples include the development of innovative food ingredients, e.g., isomaltulose [4], isolation of natural food-based ingredients (beta-glucan [5], dietary fiber, and polyphenol [6]) and enzyme inhibitors particularly for amylase and glucosidase [7].

For several years, dietary fiber has been recognized for its beneficial contribution to overall human health [8]. Dietary fiber may be described as a carbohydrate that is not broken down in the upper intestinal tract [9] but often fermented in the colon. Dietary fiber can be classified into two primary classes: soluble and insoluble. Both exhibit unique structural characteristics and therefore have differing physiological effects [10]. Soluble fiber has been linked to lowering cholesterol [11] and decreasing intestinal absorption of glucose [12]. In contrast, insoluble dietary fiber promotes the absorption of water; improves stool volume and transit time; and has a regulatory influence on intestinal absorption [13]. While structural differences in fibers may drive physiological differences, in some cases, soluble and insoluble fibers have similar effects such as contributing to stool bulking. Furthermore, although traditional sources of fibers like whole grains, fruits, and vegetables are important, added fibers, such as foods with beta-glucan, are also important contributors to dietary fiber intake [14].

Epidemiological evidence of the health benefits of dietary fiber has prompted both the health and food sectors to incorporate dietary fibers into a variety of food products. Despite a range of studies on the health benefits of dietary fiber, considerable research has emerged in the past decade or so [15,16,17,18] on the health benefits of soluble corn fiber (SCF). SCF have also been shown to have higher digestive tolerance [19].

SCF is a glucose polymer obtained from partially hydrolysed corn starch and contains a mixture of α 1-6, α 1-4, and α 1-2 glucosidic linkages. It is a soluble prebiotic fiber that has shown to increase short-chain fatty acid production and beneficial bacteria concentration in the gastrointestinal tract while decreasing end-products of protein fermentation in vitro [16,20]. These metabolic outcomes play a vital role in gut health. Previous randomized controlled trials reported increased calcium absorption and calcium retention amongst participants who added SCF in their habitual diet [18,21].

SCF has also been previously shown to elicit a low post-prandial blood glycemic response (GR) and insulin response (IR) compared to glucose control [22]. The SCF may decrease intestinal transit time, leading to more gradual nutrient absorption and prolonged feeling of satiety [22]. The slower nutrient uptake may have an effect on reducing postprandial GR and IR. The lower glucose and insulin concentration may in turn lead to reduced inhibition of lipolysis, higher circulation of nonessential fatty acid concentration and thus increase fat oxidation [22]. However, there are limited studies on SCF being incorporated into commonly consumed food products such as rice. Rice is a staple food commonly consumed in Asia. However, it is also a high glycemic index (GI) food. Hence, investigation into the glycemic and insulin effect of consuming rice with fiber will be of interest in Asia especially if it could possibly lead to beneficial effects. Hence, the objective of this study is (i) to determine the GR and IR of foods with SCF, maltodextrin (polysaccharide) and glucose (monosaccharide), and (ii) to compare the effect of replacing 50% of total carbohydrate with maltodextrin and soluble fibers such as SCF.

## 2. Materials and Methods

### 2.1. Participants

Chinese males aged between 21–60 years with body mass index (BMI) between 18.5–30.0 kg/m^2^ and fasting blood glucose below 6 mmol/L were recruited. Participants with any of the following were excluded: presence of metabolic diseases (such as diabetes, hypertension, and so on), glucose-6-phosphate dehydrogenase deficiency (G6PD deficiency), medical conditions and/or taking medications known to affect glycaemia, intolerances or allergies to foods, consumption of fibre supplements, intentional restriction of food intake, smoking, and participation in sports at the competitive and/or endurance levels. A total of 27 participants were screened, and 23 participants were recruited. Four participants failed screening due to BMI or fasting blood glucose not within range. One participant withdrew from the study due to time commitment issues. Hence, the final analysis was done on 22 participants. The study was conducted in accordance with the guidelines laid down in the Declaration of Helsinki, and all procedures involving human participants were approved by the Domain Specific Review Board (DSRB) of National Healthcare Group, Singapore (Reference no. 2017/00428) and registered in clinicaltrials.gov (Identifier: NCT03279107). Participants gave their informed consent prior to their participation.

### 2.2. Study Protocol

The study was conducted using a randomized, crossover design with seven test sessions separated by five-day washout period. Before each test session, participants consumed a standard dinner consisting of Charoen Pokphand (CP) teriyaki chicken with rice (CPF (Thailand) Public Co., Ltd., Bangkok, Thailand) and a packet of Jack N’ Jill Dewberry blueberry flavoured sandwich cookies (URC (Thailand) Co., Ltd., Bangkok, Thailand). Given that body weight predicts one’s basal metabolic rate (BMR) and total energy requirements can be predicted by BMR [23], participants with a higher body mass index (BMI) (BMI > 23.0) (calculated by dividing body weight (kg) by height (m) square) were provided with an additional Milo-malt chocolate beverage (Nestlé Singapore (Pte) Ltd., Singapore, Singapore) in order to meet their energy requirement. They were asked to refrain from consuming any other food except water and to fast for 10–12 h. They also had to avoid any strenuous physical activity and consumption of alcoholic beverages the day before the test sessions.

Participants arrived at the centre around 8:30 am following an overnight fast. Weight and body fat percentage were measured using the bioelectrical impedance analysis (BIA) device (Tanita BC-418, Tokyo, Japan), and fasting blood glucose was measured via finger prick to confirm overnight fasting. Thereafter, a blood catheter was inserted in the antecubital vein of the upper arm, which was kept patent by flushing with 3 mL non-heparinised saline. Participants were required to rest for 15 min, before a baseline blood sample (6 mL) was taken. Immediately after that, participants were served a test meal, which they were required to consume within 10 min. Blood samples (6 mL each) were collected into an Ethylenediaminetetraacetic acid vacutainer (EDTA-vacutainer) at every 15 min for the first hour and every 30 min for the next one hour upon consumption of test meal. Participants were encouraged to remain seated during the test period. The blood samples were centrifuged at 1500× *g* for 10 min at 4 °C, and the supernatant plasma was then aliquoted and stored at −80 °C for further analysis. Plasma samples were sent to National Referral Laboratories Pte Ltd., Singapore to measure the insulin and glucose concentration (µU/mL).

### 2.3. Test Meals

For the first three test sessions, 50 g glucose anhydrous powder dissolved in 250 mL water was served (Full Glucose (FG)), followed by four different test meals containing SCF (Promitor^®^ Soluble corn fiber 90, Tate & Lyle, Prairie Stone Pkwy, Hoffman Estates, IL United States; total fiber content of 91.2%, total carbohydrates of 94.8%) or maltodextrin (Star-Dri 15 maltodextrin, Tate & Lyle, Prairie Stone Pkwy, Hoffman Estates, IL United States; total fiber content of 3.1%, total carbohydrates of 93.6%) in randomized order (using https://www.randomizer.org/). For the four different test meals (two solid and two beverages), 25 g total carbohydrates would consist of either glucose or glutinous rice and the other 25 g total carbohydrates would consist of either SCF or maltodextrin. The samples were matched for total carbohydrates instead of total available carbohydrates for a more practical application. Total carbohydrates in 26 g of SCF, 27 g of maltodextrin and 32 g of glutinous rice is 25 g each. This was based on nutrient analyses results for all test foods conducted by Medallion Laboratories, USA served in portions containing 50 g of total carbohydrates (Table 1). The four test meals consisted of 32 g glutinous rice + 27 g of maltodextrin (Maltodextrin with rice (M-rice)) or 32 g glutinous rice + 26 g SCF (SCF with rice (S-rice)) and two beverages: 25 g glucose + 27 g of maltodextrin (Maltodextrin-Replaced Drink (MRD)) or 25 g glucose + 26 g SCF (SCF-Replaced Drink (SRD)). The two rice meals were prepared by steaming glutinous rice with 38 mL of hot water for 20, min and SCF/maltodextrin was mixed into the rice with 5 mL of water. The mixture was steamed for an extra 5 min before serving. The two beverages were prepared by dissolving the SCF/maltodextrin + glucose powder in 250 mL of water. All test meals were prepared in the morning of the study and were served with 250 mL water.

### 2.4. Statistical Analysis

A previous study tested the glycemic and insulin response of healthy adults to soluble corn fiber using a total replacement method, i.e., 25 g glucose replaced with 25 g soluble corn fibre. In that study, only 12 participants were sufficient to detect significant reductions in postprandial glucose and insulin [17]. Due to different study design used in this current study, i.e., only partial replacement of glucose with soluble corn fibre, the effects are predicted to be smaller, and we hence anticipate that 19 participants will allow us to detect a clinically meaningful effect of *d* = 0.7 (medium to large effect size) using a crossover study design at 80% statistical power and alpha = 0.05. Twenty-five participants were recruited to allow for 25% of attrition rate.

All statistical analyses were done on the incremental values. Incremental values were obtained by taking the difference of blood glucose readings at each time point from the fasting baseline value. The incremental area under the curves for glucose (iAU*G*C) and insulin (iAU*I*C) were calculated using the trapezoidal rule over 130 min, ignoring the area beneath the baseline [24]. Log transformation was undertaken to achieve normal distribution where necessary. The iAU*G*C and iAU*I*C were analysed using repeated measures ANOVA with Tukey HSD correction for the post hoc tests comparing all the test foods (Bonferroni correction was used for the post hoc comparison between the beverage treatments). Incremental peak plasma glucose values were compared between the beverage treatments using repeated measures ANOVA and Bonferroni correction. Significant treatments × time interaction for both glucose and insulin was followed with comparisons of treatment effects at each time point using Bonferroni corrections. Relative response (RR) for glucose and insulin were calculated using Equation (1) as follows:(1)$$RR=\frac{iAUC\;of\;test\;meal}{iAUC\;of\;50g\;of\;glucose\;drink}$$

All statistical analysis in this study were done using SPSS version 24.0 (IBM Corp, Armnok, NY, USA). Statistical significance was set at *p* < 0.05, two-tailed.

## 3. Results

The anthropometric characteristics of the study participants in this study is presented in Table 2.

### 3.1. Blood Glucose Response

The mean iAU*G*C (130) of both SRD and S-rice were significantly lower than the mean iAUGC (130) of the rest of the treatments (*p* < 0.05), as seen in Figure 1A. The mean iAU*G*C (130) of MRD and M-rice were significantly higher than the SCF treatment (*p* < 0.05) but not significantly different from the FG treatment (*p* > 0.05). Figure 1 also shows that the plasma glucose concentration at each time point was significantly lower than of FG, indicating a gradual increase and decrease.

This pattern was also observed in the comparison made between the beverages, i.e., FG, SRD, and MRD. The mean iAU*G*C (130) of SRD was significantly lower than the rest of the two treatments (*p* < 0.01). Similarly, the mean incremental peak glucose values after the consumption of the SRD was significantly lower than the MRD (*p* = 0.03).

Moreover, the postprandial glucose levels of SRD were significantly lower (*p* < 0.05) than the FG and MRD between 55 min and 100 min (Figure 2A). In addition, the RR of glucose for MRD was significantly higher than the RR of SRD (*p* < 0.001).

### 3.2. Blood Insulin Response

The mean iAU*I*C (130) of S-rice was significantly lower than the rest of the treatments, including the treatments that have been replaced with maltodextrin (*p* < 0.05). M-rice and MRD treatments resulted in similar mean iAU*I*C (130) values as the FG (*p* > 0.05), but there was moderate statistically significant difference between MRD and FG (*p* = 0.05).

Similarly, the mean iAU*I*C (130) of the SRD was significantly lower than the rest of the beverages (*p* < 0.05). However, the mean iAU*I*C (130) of MRD was significantly higher than the FG (*p* = 0.005) when comparing only among the beverages. The postprandial insulin levels of the SRD treatment were significantly different from the rest of the treatments between 40 min and 130 min (*p* < 0.05, Figure 2B). In addition, the RR of insulin for MRD was significantly higher than that of SRD (*p* < 0.001).

## 4. Discussion

Faced with an epidemic of both prediabetes and T2DM, strategies to prolong the transition from prediabetes to T2DM have become a major research focus. Lifestyle interventions such as following a healthier diet may delay the progression for as long as 10 years [25,26]. This has encouraged more research on dietary fibers as an ingredient to be incorporated into the diet. The current study has compared the effect of replacing 50% of total carbohydrates with either soluble fiber or maltodextrin.

In this study, it was observed that the treatment which contained SCF resulted in significantly lower mean iAU*G*C and iAU*I*C as compared to treatments with maltodextrin (Figure 1A,C). In fact, mean iAU*I*C of S-rice was significantly lower than the rest of the treatments administered (Figure 1C). In particular, replacing 50% of total carbohydrate with SCF (SRD) had significantly lower mean iAU*G*C, iAU*I*C, and incremental peak glucose value as compared to MRD (Figure 1).

Solid foods have inherently higher gastric emptying rates as compared to liquid state foods [27]. Different gastric emptying rates could have an effect on the postprandial glycemic and insulin observations [28]. Hence, further investigations into glycemic and insulin-reducing properties of SCF and maltodextrin were done by comparing only the liquid beverages: FG, SRD, and MRD. This will also be appropriate given that there is a suitable control, FG, in the liquid state as compared to the solid state allowing for a good comparison.

SCF was seen to have beneficial impact on attenuating the blood GR as compared to maltodextrin (Figure 2A). The postprandial glucose concentration increased for FG, MRD, and SRD. However, consuming SRD resulted in the blood glucose response returning back to baseline faster compared to MRD and FG. This suggests that the addition of SCF reduced the fluctuation in the postprandial glycemic response as compared to the addition of maltodextrin. This can be seen in Figure 2A where the incremental plasma glucose concentrations of SCF was significantly lower than MRD and FG between the 50 min to 100 min interval. On the other hand, the incremental plasma glucose concentration values of the MRD and FG were very similar (Figure 2A).

Replacing 50% of total carbohydrates with SCF also attenuated the IR compared to replacement with maltodextrin in beverages. The increase in IR of FG and MRD was significantly higher than SRD (Figure 2B). The IR of FG and MRD remained higher than SRD from 40 min to 130 min (Figure 2B). Furthermore, the incremental plasma insulin concentration values of MRD and FG were very similar and different from SRD over the 40 min to 130 min duration (Figure 2B). From the RR of glucose and insulin (Figure 2), MRD had increased both the postprandial glycemia and insulin secretion by about 20% and 40% respectively. In contrast, the replacement with SCF attenuated glucose excursion as well as reduced insulin secretion by 20% and 40%, respectively. This difference in the RR of glucose and insulin between MRD and SRD was statistically significant. This reaffirmed the beneficial effect of replacing total carbohydrates with soluble fiber such as SCF.

The inclusion of SCF had better postprandial GR and IR as compared to maltodextrin or FG. This observation may be attributable to SCF’s ability to resist hydrolytic digestion and absorption in the small intestine. A study in pigs reported that at least 70% of SCF resisted digestion and were passed into the large intestine for fermentation [29]. The position of the predominant glycosidic linkages is known to contribute to the low digestibility of SCF. Kendall CW et al. [17] evaluated the in vitro digestibility of SCF, and it was reported to have a digestibility of 14.5%. Ropert*,* et al. [30] suggested that the short-chain fatty acids produced by colonic fermentation of carbohydrates may also be a meditator of gastric motility. The organic acid and especially short-chain fatty acids can also acutely reduce the iAU*G*C. The significantly lower incremental peak GR and iAU*G*C (130) observed in SRD may be of interest when formulating foods that produce a lower postprandial GR. However, longer term studies are necessary to understand and confirm these short-term observations.

Despite maltodextrin being classified as a long-chain carbohydrate, there were no differences observed between the glucose (simple sugar) treatment and maltodextrin in relation to their iAU*G*C, as seen in Figure 2A (*p* = 0.57). The glucose and insulin postprandial response of MRD was similar to that of FG (Figure 2). Although maltodextrin has higher polymeric complexity than glucose, the weak bonds could have resulted in high digestibility which will be rapidly absorbed in the gut like glucose [31].

There are a few limitations to our study. The lack of a representative solid form of FG in the study design restricts the analysis to FG, SRD, and MRD. Hence, this only allows the investigation of the effect of replacing available carbohydrates with soluble fiber and maltodextrin in beverages. Secondly, the total available carbohydrate in SCF was not measured. Thirdly, this is an acute study where the immediate glycemic response after consuming the meals is measured. In order to study the long-term benefits of consuming the treatment meals, a long-term trial should be conducted in the future. However, the strength of this study is that, for the first time, a head-to-head comparison between maltodextrin and SCF has been made where the primary objective was to determine the potency of SCF to attenuate postprandial glycaemia. Such studies should provide further insights in the use and application of SCF as a useful ingredient in the development of various low GI foods. Consumers of today are demanding all over the world for a variety of palatable low GI meals and snacks. Our study points to the potential application of SCF in these products.

## 5. Conclusions

In conclusion, foods incorporated with SCF had lower mean iAUGC and iAUIC as compared to foods with maltodextrin. SRD had significantly lower mean incremental peak glucose values, resulting in lower iAUGC as compared to FG and MRD. Hence, the results show that the inclusion of SCF into the diet is beneficial in controlling the postprandial glycemic profile. Replacing total carbohydrates with SCF lowers GR and IR and may possibly delay the transition of subjects from prediabetes to diabetes. This study prompts future exploration in incorporating SCF into commonly consumed foods in an attempt to reduce the glycemic response as well as to investigate the effects of prolonged consumption of such foods on the transition from prediabetes to diabetes.

## Figures and Tables

**Figure 1 nutrients-12-00961-f001:**
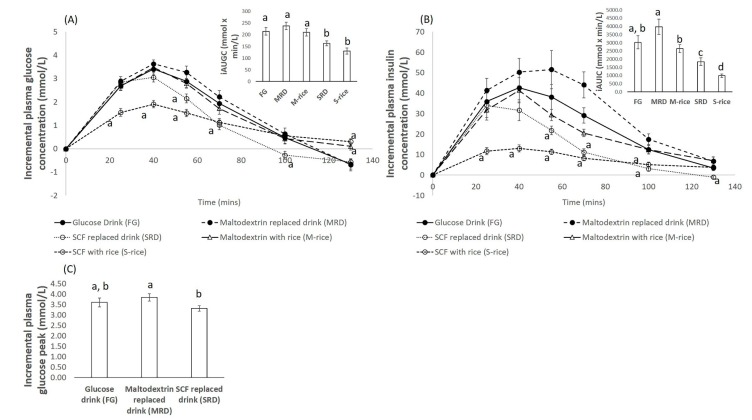
(**A**) Incremental plasma glucose concentration over time of all the five treatments along with the bar plot of the mean incremental area under the curve values of glucose (iAUGC) (130) of the five different treatments: The error bars represent the standard error of the mean (SEM). Each time point was compared with the glucose drink as the reference. (**B**) Incremental plasma insulin concentration over time of all the five treatments along with the bar plot of the mean iAU*I*C (130) of the five different treatments: The error bars represent the SEM. Each time point was compared with the glucose drink as the reference. (**C**) Bar plot of the mean incremental plasma glucose peak of the three liquid treatments. The error bars indicate the SEM. For all plots, different alphabets represent statistically different mean values.

**Figure 2 nutrients-12-00961-f002:**
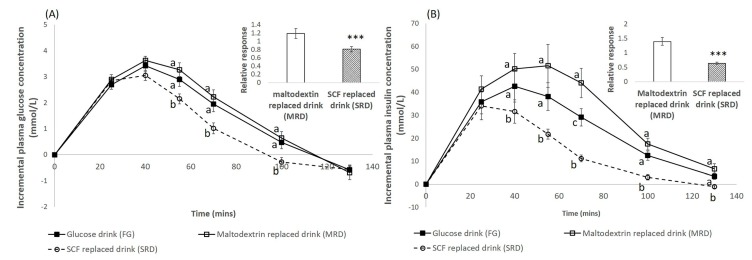
(**A**) Incremental plasma glucose concentration over time of the three beverage treatments along with the bar plot showing the mean relative response of maltodextrin replaced drink (MRD) and SCF-replaced drink (SRD): Relative response is calculated by dividing iAU*G*C of the test meal by iAU*G*C of the glucose drink. The error bars represent SEM. (**B**) Incremental plasma insulin concentration over time of the three beverage treatments along with the bar plot showing the mean relative response of maltodextrin replaced drink (MRD) and SCF-replaced drink (SRD): Relative response is calculated by dividing iAU*I*C of the test meal by iAU*I*C of the glucose drink. The error bars represent SEM. For all plots, different alphabets represent statistically different mean values. *** *p* value < 0.001. Mean relative responses of MRD and SRD were compared using paired sample *t* test.

**Table 1 nutrients-12-00961-t001:** Serving portion sizes and nutrient composition of four test foods (per serving of 50 g total carbohydrate product).

Test Food	Serving Size (g)	Energy (kJ)	Energy (kcal)	Fat (g)	Protein (g)	Dietary Fiber (g)
SCF ^1^ glutinous rice (S-rice)	101	883	211	0.2	2	25
Maltodextrin glutinous rice (M-rice)	101	883	211	0.2	2	1.8
SCF beverage (SRD)	282	812	194	0	0	24.1
Maltodextrin beverage (MRD)	282	812	194	0	0	0.8

^1^ SCF: Soluble Corn Fibre.

**Table 2 nutrients-12-00961-t002:** This is a table displaying the anthropometric characteristics of the study participants analysed in this study (*n* = 22).

Anthropometric and Physiological Parameters	Mean ± SEM	Range
Age (years)	36.7 ± 2.8	23.8–60.8
Height (cm)	172.1 ± 1.5	158.4–181.1
Weight (kg)	70.2 ± 1.7	52.5–78.9
BMI (kg/m^2^)	23.6 ± 0.3	20.3–26.7
Waist circumference (cm)	83.0 ± 1.4	71.1–94.1
Hip circumference (cm)	96.4 ± 1.3	79.0–105.8
Fasting blood glucose (mmol/L)	4.9 ± 0.1	4.0–5.7
Systolic Blood pressure (mmHg)	125.1 ± 2.1	102.0–146.0
Diastolic Blood pressure (mmHg)	79.5 ± 2.1	58.0–100.0

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
