# Peer review of "The Role of Soluble Corn Fiber on Glycemic and Insulin Response"

_nutrients, 2020, doi:10.3390/nu12040961_

Round 1

Reviewer 1 Report

I commend your research efforts in investigating the use of soluble corn fiber for controlling the postprandial glycemic and insulin response.

I do have a few editorial comments.  First, please change the number of subjects listed in the abstract (line 18-19) to twenty-two.  Second, please add more to the single sentence paragraph in the introduction (line 51-52), such as more detail on who are the "added fibers".  Third, on line 55 it states "in the past decade", however the cited references 15-18 span more than one decade.  Finally, please list only the first author followed by et al. on line 243-44 in the discussion.

Reviewer 2 Report

Summary

In this manuscript, Kimberly Tan et al. addressed the beneficial effect of soluble corn fiber (SCF) on glucose homeostasis and insulin response. The SCF replacement drove the recipients blood glucose and insulin levels lower, suggesting improvement of glucose response and insulin sensitivity.

It’s, as a part of nutritional study to improve glucose response, beneficial. However, there are some concerns as listed below.

Comments

1, What was the mechanism to improve glucose and insulin response? Does this simply suppress sugar absorption from the gut into the blood? Are there any data to prove it based on this study (not based on the previous study and citation)?

2, It seems the test is checking the acute reaction for one shot of study meals. How was the long-term effect? Type 2 diabetes or obesity prevention needs to long-term treatment to improve their glucose homeostasis, insulin sensitivity and QOL.

3, It is more interesting to address long-term treatment and test gut microbiota population and fecal (fecal) contents by using metabolomics.

4, Is there any difference between figure 1 and figure 2 data acquirement? If not, then are there any specific reason not to show the time course line graph for figure 1? Since authors stated the interest of SCF with rice at the end of introduction, and there is SCF with rice in figure 1, that’s more important to show time course and AUC than the replacement drinks as like figure 2.

5, The characters on the figure are too small. It is too hard to read (especially figure 2). Please use appropriate size of fonts.

6, The legend in figure 2A and 2B are inconsistent. Please fix them.

7, One of the authors seems to be an employee of the company providing SCF. If this paper published, it may potentially help to his/her company making profits. It’s better to indicate it in addition to the declaration sentence.

8, What are the components of the SCF and can authors provide the structure for SCF?

9, The authors concluded that “the replacement of total carbohydrate with SCF lowers…”. It might be true. However, is it possible to do so? And is it possible to continue for long span? One day effect does not help a lot to “delay the transition of subjects from prediabetes to diabetes” as the authors concluded. It must be continuously doable. Therefore, I recommend authors to modify the conclusion appropriately.

Round 2

Reviewer 2 Report

In this manuscript, Kimberly Tan et al. addressed the beneficial effect of soluble corn fiber (SCF) on glucose homeostasis and insulin response. The SCF replacement drove the recipients blood glucose and insulin levels lower, suggesting improvement of glucose response and insulin sensitivity.

It’s, as a part of nutritional study to improve glucose response, beneficial.

The authors addressed most of my comments generously. Here I commented to their response. Please check them.

Point 1:

Response to authors: The explanation is how maltodextrin increases postprandial glycemic response. I agree to their explanation about the postprandial glycemic response. However, I believe this study is on addressing the beneficial effect of SCF treatment. Therefore, it is required to explain how SCF reduces postprandial glycemic response with “reasonable data”.

Point 2:

Response to authors: OK

Point 3:

Response to authors:OK

Point 4: 

Response to authors: OK

Point 5:

Response to authors: I could see the improvement. However, related to point 6, even authors couldn’t find out the difference between two legends on the Y axis of bar graphs, it might be still too small. I suggest authors to indicate only “Relative response” on the figures and indicate the following explanation in the figure legend part below figures.

Point 6:

Response to authors: In the figure 2, there are bar graphs on the right shoulder for both A and B. Figure 2A is “Relative response (iAUGC test meal/iAUGC of glucose drink)” and 2B is “Relative response (iAUGC “OF” test meal/iAUGC of glucose drink)”. This difference is still on the revised version. Is there any specific reason to keep that? Please explain.

Point 7:

Response to authors: OK. And I’ll pass the decision of justification for editor and journal policy.

Point 8:

Response to authors: OK

Point 9: 

Response to authors: OK
